# Effect of Ultra-Processed Foods Consumption and Some Lifestyle Factors during Pregnancy on Baby’s Anthropometric Measurements at Birth

**DOI:** 10.3390/nu15010044

**Published:** 2022-12-22

**Authors:** Rafaela Cristina Vieira e Souza, Cristianny Miranda, Taciana Maia de Sousa, Luana Caroline dos Santos

**Affiliations:** Department of Nutrition, School of Nursing, Federal University of Minas Gerais, Belo Horizonte 30140-100, MG, Brazil

**Keywords:** ultra-processed foods, pregnancy, birth weight, anthropometric measurements, food consumption

## Abstract

Objective: This study aimed to assess the association between ultra-processed foods (UPF) consumption and some lifestyle factors during pregnancy and the baby’s anthropometric measurements at birth. Methods: Cross-sectional study conducted with immediate postpartum women and their babies. Food consumption during pregnancy was assessed retrospectively by a semi-quantitative food frequency questionnaire, and the baby’s anthropometric measurements were obtained from the medical records. The percentual of energy from UPF was estimated, categorized in tertiles, and associated with the baby’s anthropometric measurements using multinomial logistic regression, adjusting by maternal characteristics (per capita income, maternal age, gestational weight gain, parity, physical activity, and number of prenatal consultations). Results: A total of 626 immediate postpartum women and their babies were evaluated. The mean percentual of energy from UPF consumption during pregnancy was 30.56%. Before adjustment, there was a greater chance of insufficient birth weight among babies of immediate postpartum women in the highest tertile of UPF consumption (OR 1.72; 95% CI 1.09–2.70; *p* = 0.020); however, such an association was not significant in the adjusted analyses. Conclusions: No association between UPF consumption during pregnancy and baby’s anthropometric measurements at birth was observed, probably due to the multifactorial nature of anthropometry and the interference of sociodemographic, gestational, and environmental factors in the baby’s health outcomes.

## 1. Introduction

Ultra-processed foods (UPF) are products that generally do not maintain their basic characteristics for undergoing different processes and techniques to have industry-exclusive substances included in their composition [1]. UPF are nutritionally deficient, because they tend to be rich in sugar, saturated and trans-fat, and sodium, with high energetic density, and they are low in protein, fibres, and important micronutrients for fetal growth [1,2]. Therefore, this consumption can lead to alterations in children’s lipidic profile, metabolic syndrome in teenagers, and obesity in adolescents and adults [1].

During pregnancy, the intake of UPF must be moderate, since an inadequate intake of macro and micronutrients can alter maternal gestational weight gain (GWG) and blood pressure and glycemia, and as a consequence, intrauterine growth, which leads to offspring with birth weight and length deviations [3,4]. Even with the tendency of women decreasing ultra-processed food consumption and adopting a healthier pattern during pregnancy, the contribution of UPF to the pregnant woman’s diet remains high, representing about a third of the total daily energy consumed, which contributes to the increase in the risk of complications for the mother and baby [3].

Low birth weight (LBW; <2500 g) is associated with a higher neonatal and infant morbimortality, being the most influential factor in the first years of life [5]. Babies with insufficient weight at birth (2500 g to 2999 g) have up to three times the chances of dying than those with adequate weight, besides having a higher chance of inadequate growth and development [6]. On the other hand, fetal macrosomia (>4000 g) is associated with neonatal asphyxia and higher chances of fetal hypoglycemia [7].

Little is known about the association between diet quality and another factors, such as physical activity and parity during pregnancy, and birth weight [3,5]. Moreover, the influence of UPF consumption during pregnancy on measurements deviations at birth remains uncertain [8], and up to this moment, there are no studies that used NOVA classification to clarify the role of UPF in this association.

Considering the importance of the baby’s anthropometric measurements on child health and the global increase of UPF consumption in the last decades, this study aimed to assess the influence of UPF consumption and some lifestyle factors during pregnancy on the baby’s anthropometric measurements at birth.

## 2. Materials and Methods

This cross-sectional study evaluated immediate postpartum women and their babies in a Brazilian public maternity hospital between 2018 and 2019.

For sample calculation, the prevalence of birth weight for gestational age > 90th percentile among mothers with high adherence to a dietary pattern based on fast food and candies (7%) found in a similar study were adopted [4], so the minimum sample size was 138 participants, with a significance level of 5% and a statistical power of 95%.

The exclusion criteria were based on variables that could interfere with the baby’s anthropometric measurements [9]: diabetes, birth complications (including severe hypertension (eclampsia and preeclampsia)), HIV infection (human immunodeficiency virus infection), and multiple gestations.

This study was conducted in accordance with the Helsinki Declaration, and all procedures involving human participants were approved by the Ethical Committee of Universidade Federal de Minas Gerais and Fundação Hospitalar de Minas Gerais. Written informed consent was obtained from all immediate postpartum women.

Maternal level of education was categorized as primary education, high school, or higher education. Per capita income was categorized in <0.5 and ≥0.5 minimum wage (considering the average of Brazilian minimum wage in 2018 and 2019: R$954.00 and R$998.00, respectively). Maternal marital status was classified as married/civil union or single/divorced/widow.

Pre-gestational body mass index (BMI) was calculated using the self-reported pre-gestational weight and height was measured by the researchers according to the World Health Organization (WHO) [10] recommendation, using a Cescorf^®^ portable stadiometer. Z-scores cutoff points of BMI/age [11] were obtained with the software AnthroPlus^®^ and were used for the adolescents (<20 years old). The WHO8 recommendation was considered for adult mothers. The categories “overweight” and “obesity” were included in the category “excessive weight”.

If the mothers could not inform the GWG, the weight registered in the pregnant woman diary from the last prenatal consultation was considered for calculation. Institute of Medicine [12] criteria were used to classify the GWG as insufficient, adequate, or excessive.

Parity was categorized as primiparous and multiparous. The number of prenatal consultations was classified as not attended, one to five, and six or more consultations, and the last one covers the Brazilian recommendation of a minimum of six consultations during pregnancy [13].

Smoking practice was classified as smoker, ex-smoker, or never smoked. The practice of pre-gestational physical activity was classified according to the level of physical activity criteria [14]. Women included in the “sedentary” and “low active, active and very active” categories were classified as “sedentary” and “non-sedentary”, respectively.

Food consumption during pregnancy was measured using a semi-quantitative food frequency questionnaire (FFQ), validated for the Brazilian adult population [15]. The questionnaire consisted of 57 items, with their respective frequencies of food consumption during pregnancy (last nine months) and quantifications expressed in homemade measurements. Data was converted into daily intake using the appropriate tables [16,17,18,19,20]. The energy composition was computed using the Excel program inserting appropriate food tables and their respective compositions [16,17,18,19,20], in addition to food labels, when necessary.

The items in the FFQ were classified according to the NOVA classification system, into four distinct groups according to Monteiro et al.’s recommendations [21,22]: in natura or minimally processed foods, processed food, and UPF. The UPF group was composed by: carbonated soft drinks, sweet or savory packaged snacks, chocolate, candies, ice-cream, mass-produced packaged breads and buns, margarine, mayonnaise, cookie (biscuit), cake mixes, yogurts, ‘fruit’ drinks, ‘instant’ sauces, artificial juice, ‘instant’ noodles, sausages, burgers, and hot dogs.

The results of each group consumption frequency were converted into energy (Kcal/day) for further classification into tertiles referring to the percentage of the total daily caloric value from these groups of foods (% of total energy content—TEC) [23]. Participants who had implausible energy intake, <300 Kcal/day or >10,000 Kcal/day [24] were excluded from database (n = 6).

Baby’s anthropometric measurements were obtained from the medical records. Birth weight was classified as low <2500 g; insufficient: 2500 to 2999 g; adequate: 3000 g to 4000 g; and macrosomia: ≥4000 g [10]. Birth weight, according to gestational age, was classified on: <10th percentile: small for gestational age (SGA); >10th percentile and <90th percentile: adequate for gestational age (AGA); and >90th percentile: large for gestational age (LGA), according to the INTERGROWTH-21st [25]. Birth length was evaluated according to the cutoff points proposed by WHO [26].

Data were double-typed using the EpiInfo^TM^ program, and after consistency analysis, descriptive analysis was performed. Kolmogorov–Smirnov test was applied, and Chi-square or Fisher’s exact test with Bonferroni post-hoc were performed. These analyses were performed using the Statistical Package for the Social Sciences (SPSS) version 21.0.

Multinomial logistic regression was performed using Stata^®^ program, version 14.2, to verify the associations between baby’ anthropometric measurements at birth and the UPF tertiles, adjusting by important maternal characteristics (per capita income, maternal age, gestational weight gain, parity, physical activity, and number of prenatal consultations), selected considering its potential interference on the outcome [9]. The significance level of 5% was adopted for all analyses.

## 3. Results

A total of 626 immediate post-partum women and their babies were evaluated (Appendix A). The median maternal age was 26 (18–44) years old. A high prevalence of mothers with insufficient (31.7%) or excessive (29.2%) GWG was observed. Regarding supplementation, 88.5% of women used during pregnancy, with the main ones reported being folic acid, iron, and multivitamins. Additional maternal characteristics during pregnancy are described in Table 1.

The mean energy percentual from UPF consumption during pregnancy was 30.56% (Appendix A). Mothers in the highest tertile of UPF consumption were more likely to be classified in the lowest tertile of in natura and minimally processed foods consumption (OR 5.08 95% CI 3.17–8.14).

As for the babies, 11.4% were SGA, and 13.3% presented small length at birth. Additionally, 6.9% were LBW, 25.2% presented insufficient birth weight, and 3.5% presented macrosomia (Table 1).

In the bivariate analysis, marital status, pre-gestational BMI, GWG, prenatal consultations, and smoking were associated with the baby’s weight at birth. The variables GWG and parity were associated with weight for gestational age at birth. Additionally, GWG was associated with length at birth. No association was observed between the UPF tertiles and the baby’s anthropometric measurements at birth (Table 1).

In the regression model, before adjustment, there was a greater chance of insufficient weight at birth among babies of mothers in the highest tertile of UPF consumption. However, such association was not observed after adjustments (Table 2).

## 4. Discussion

This research found no association between the consumption of UPF during pregnancy and the baby’s anthropometric outcomes at birth, probably due to the multifactorial nature of anthropometry and the interference of sociodemographic, gestational, and environmental factors.

Similar to LBW, insufficient weight at birth represents an important risk factor for infectious diseases, such as diarrhea, acute respiratory infections, delayed growth and development, high infant mortality rate, and future chronic diseases. The frequency of offspring born with insufficient weight is considerably higher than LBW, as observed in the present study, and both conditions can be determined by several socioeconomic, environmental, and behavioral factors [27].

Such factors are described as having a strong influence on anthropometric measurements at birth, with repercussions in childhood and adulthood [2,9], which may justify the fact that, after the adjustment, the chance of birth of babies with insufficient weight among mothers with higher consumption of UPF lost the statistical significance. Factors such as low level of education and maternal income, age under 20 or over 35 years [28], primiparity, <6 prenatal consultations [8], insufficient GWG [29], among others, are determinants for low or insufficient birth weight.

In this research, GWG was associated with all the evaluated outcomes in the bivariate analysis, corroborating with other findings. A study by Santana et al. [30], carried out in a Brazilian municipality, observed that women with inadequate GWG had a 2.6 times greater chance (95% CI 1.5–3.5) of having offspring with inadequate birth weight (≤2999 g and ≥4000 g), when compared to women with adequate GWG.

Although the association between the UPF consumption and the insufficient weight at birth has lost statistical significance after adjustment for confounding variables, maternal food consumption stands out in the literature among the explanatory variables for baby’s anthropometric outcomes, for the diet of pregnant woman must contain adequate energy and nutrients to allow healthy fetal growth and development [28]. Changes in eating habits, such as increasing consumption of in natura and minimally processed foods (as fruits and vegetables) and reducing the intake of UPF (as sugary drinks and others), should be encouraged by health professionals during prenatal care [29].

Another point that deserves attention and could explain the absence of associations from UPF intake and outcomes is that the underreporting of food intake could occurs when the data are collected using FFQ [31], despite a validated questionnaire was used. Moreover, the declared food intake may also not correspond to the nutritional status measured by concentration in blood (blood biomarkers), as a result of different bioavailability of each nutrient and individual differences in metabolism [32].

In the present study, the mean energy percentual from UPF (30.6%) was similar to that found in other studies conducted with pregnant women. In a cross-sectional study carried out in Brazil with 785 women, an average contribution of 32% of the TEC from UPF was identified [3].

There is a gap in the literature regarding the acceptable level of UPF consumption. Crivellenti et al. [33], in the study that developed the adapted diet quality index for pregnant women, proposed the cutoff point of 18% for the consumption of UPF, a value that corresponded to the 16th percentile of the sample used. However, this cutoff point has not been validated to be used isolated [34]. Thus, we do not classify the mean energy percentual from UPF as adequate or excessive, but strategies to promote healthy eating among pregnant women are important and urgent.

Studies that demonstrate the risk of UPF consumption during pregnancy for the baby’s anthropometric measurements in the short and long term are still scarce. However, it is known that the consumption of these products can impair the quality of food intake, due to their ability to displace and interfere with the consumption of foods that are beneficial to health, such as in natura and minimally processed foods [35]. The present study demonstrated this interference, considering the inverse association observed between the consumption of in natura and minimally processed foods and UPF.

Studies carried out in Brazil and worldwide have shown controversial results on the association between the consumption of UPF during pregnancy and the baby’s anthropometric measurements [4,36].

In a Brazilian prospective cohort, the pattern composed of UPF, such as fast food and snacks, cakes, cookies, candies, or desserts, was associated with a greater chance for LGA babies and birth length > 90th percentile [4]. On the other hand, a recent systematic review [35] showed that there is no influence of UPF consumption during pregnancy on the baby’s anthropometric measurements up to one year of life.

Maternal self-reported pre-gestational weight and GWG may be influenced by memory and underestimation bias and is, therefore, considered a limitation of this study. However, the literature demonstrates a good agreement and validity between the reported and measured weight [37].

Finally, it is worth noting that the NOVA classification is relatively recent, and the authors are still unaware of studies using this new methodology to investigate the association between UPF consumption during pregnancy and the baby’s anthropometric measurements at birth. 

## 5. Conclusions

The consumption of UPF during pregnancy was not associated with the baby’s anthropometric measurements at birth, probably due to the multifactorial nature of anthropometry and the interference of sociodemographic, gestational, and environmental factors in the outcomes.

## Figures and Tables

**Table 1 nutrients-15-00044-t001:** Anthropometric measurements of the baby according to socioeconomic and demographic, gestational, prenatal, and maternal lifestyle characteristics and the tertiles of ultra-processed foods consumption during pregnancy (2018–2019).

Variable	Total n (%)	Weight at Birth				*p* Value ^a^	Weight for Gestational Age at Birth			*p* Value ^a^	Length at Birth			*p* Value ^a^
		LW 43 (6.9)	IW 157 (25.2)	AW 400 (64.3)	MC 22 (3.5)		SGA 70 (11.4)	AGA 494 (80.6)	LGA 49 (8.0)		SBL 81 (13.3)	NBL 525 (85.9)	LBL 5 (0.8)	
		(%)	(%)	(%)	(%)		(%)	(%)	(%)		(%)	(%)	(%)	
**Education**														
Primary education	103 (16.5)	9.8	25.5	60.8	3.9	0.579	11.2	81.6	7.1	0.626	13.4	85.6	1.0	0.983
High school	412 (65.9)	5.6	24.4	66.6	3.4		10.6	80.5	8.9		12.8	86.4	0.7	
Higher education	110 (17.6)	9.2	28.4	58.7	3.7		14.7	79.8	5.5		14.8	84.3	0.9	
**Per capita income (minimum wage)**														
<0.5	245 (42.9)	5.3	27.9	63.1	3.7	0.348	10.4	81.3	8.3	0.902	16.2	82.9	0.9	0.078
≥0.5	326 (57.1)	8.0	22.4	66.0	3.7		11.5	79.9	8.7		9.9	89.5	0.6	
**Maternal age (years)**														
≤19	59 (9.4)	8.8	26.3	61.4	3.5	0.996	18.5	77.8	3.7	0.327	22.4	75.9	1.7	0.173
20 to 35	500 (79.9)	6.6	25.3	64.5	3.6		10.4	81.1	8.5		11.9	87.2	0.8	
≥36	67 (10.7)	7.5	23.9	65.7	3.0		13.4	79.1	7.5		14.9	85.1	0.0	
**Marital status**														
Married/civil union	285 (45.5)	4.9 ^a^	21.6 ^c^	69.3	4.2	**0.038 ***	8.6	83.1	8.3	0.142	11.4	88.2	0.4	0.227
Single/divorced/widow	341 (54.5)	8.6 ^b^	28.3 ^d^	60.2	2.9		13.7	78.5	7.8		14.8	84.0	1.2	
**Pre-gestational BMI**														
Adequate	340 (55.5)	6.8 ^a^	28.9 ^c^	60.4	3.9	0.001 *	12.0	80.4	7.5	0.160	14.2	85.2	0.6	0.265
Low weight	35 (5.7)	17.1 ^b^	40.0 ^c.d^	42.9	0.0		22.9	74.3	2.9		22.9	77.1	0.0	
Overweight	238 (38.8)	5.9 ^a.b^	18.5 ^d^	72.3	3.4		9.4	81.6	9.0		10.8	87.9	1.3	
**Gestational weight gain**														
Insufficient	182 (31.7)	14.4 ^a^	35.0 ^c.e^	47.8	2.8	**<0.001 ***	19.0 ^a^	76.0	5.0 ^c.d^	**<0.001 ***	19.1 ^a.b^	80.9	0.0	**0.002 ***
Adequate	225 (39.1)	4.0 ^b^	26.3 ^d^	67.4	2.2		9.9 ^b^	86.1	4.0 ^c^		13.8 ^a^	85.8	0.5	
Excessive	168 (29.2)	3.6 ^a.b^	14.3 ^e.c^	76.2	6.0		6.7 ^a.b^	78.8	14.5 ^d^		6.7 ^b^	90.9	2.4	
**Parity**														
Primiparous	304 (48.6)	7.7	26.7	62.3	3.3	0.714	13.6	81.3	5.1 ^a^	**0.016 ***	15.1	84.6	0.3	0.195
Multiparous	322 (51.4)	6.2	23.9	66.1	3.7		3.4	79.9	10.7 ^b^		11.5	87.2	1.3	
**Prenatal consultations**														
Not attended	3 (0.5)	0.0	33.3 ^a.b^	66.7	0.0	**0.041 ***	0.0	100.0	0.0	0.238	0.0	100.0	0.0	0.203
1 to 5	71 (11.5)	11.3	39.4 ^a^	46.5	2.8		12.9	85.7	1.4		22.1	77.9	0.0	
6 or more	546 (88.1)	6.5	23.2 ^b^	66.6	3.7		11.4	79.6	9.0		12.3	86.7	0.9	
**Smoking**														
Smoker	36 (6.9)	20.0 ^a^	40.0 ^c^	40.0	0.0	**<0.001 ***	20.0	77.1	2.9	0.446	17.6	82.4	0.0	0.820
Ex-smoker	81 (15.5)	9.9 ^a.b^	14.8 ^c.d^	71.6	3.7		9.9	82.7	7.4		10.3	88.5	1.3	
Never Smoke	407 (77.7)	4.9 ^b^	25.8 ^d^	65.1	4.2		11.1	80.7	8.1		13.0	86.0	1.0	
**Physical Activity**														
Sedentary	553 (89.8)	7.3	25.7	63.4	3.6	0.841	11.4	80.3	8.3	0.361	14.3	85.0	0.7	0.187
Non-sedentary	63 (10.2)	4.8	23.8	68.3	3.2		12.9	83.9	3.2		6.5	91.9	1.6	
**UPF consumption**														
Tertile 1 (0–22.86%)	209 (33.4)	5.3	21.4	70.4	2.9	0.261	9.9	82.2	7.9	0.147	12.2	86.8	1.0	0.947
Tertile 2 (>22.86–36.18%)	208 (33.2)	7.2	24.0	64.4	4.3		10.7	78.0	11.2		13.6	85.9	0.5	
Tertile 3 (>36.18–79.28%)	209 (33.4)	8.2	30.3	58.2	3.4		13.6	81.6	4.9		14.0	85.0	1.0	

LW, low weight; IW, insufficient weight; AW, adequate weight; MC, macrosomia; SGA, small for gestational age; AGA, adequate for gestational age; LGA, large for gestational age; SBL, short birth length; NBL, normal birth length; LBL, long birth length; BMI, body mass index; UPF, ultra-processed foods. The sequence followed by different letters between the categories differed significantly by the Bonferroni test (*p* < 0.05). Missing data on education (n = 1), per capita income (n = 55), pre-gestational BMI (n = 13), GWG (n = 61), prenatal consultations (n = 6), smoking (n = 102), and physical activity (n = 10) were observed due to failure in data collection, maternal memory bias, or refusal to provide information. ^a^
*p*-value using the chi-square test. * Statistical significance.

**Table 2 nutrients-15-00044-t002:** Unadjusted and adjusted odds ratio for ultra-processed foods consumption tertiles during pregnancy associated with the anthropometric measurements of the baby, according to logistic regression (2018–2019).

UPF Consumption	Weight for Gestational Age					
	Small for Gestational Age (SGA)				Large for Gestational Age (LGA)	
	OR (95%CI)		aOR (95%CI)		OR (95%CI)	aOR (95%CI)
Tertile 1	1.00		1.00		1.00	1.00
Tertile 2	1.14 (0.60–2.17)		0.97(0.47–1.98)		1.49 (0.76–2.93)	1.68 (0.75–3.80)
Tertile 3	1.38 (0.75–2.55)		1.36 (0.68–2.72)		0.62 (0.27–1.40)	0.77 (0.30–1.98)
	**Weight at birth**					
	**Low**		**Insufficient**		**Macrosomia**	
	**OR (95%CI)**	**aOR (95%CI)**	**OR (95%CI)**	**aOR (95%CI)**	**OR (95%CI)**	**aOR (95%CI)**
Tertile 1	1.00	1.00	1.00	1.00	1.00	1.00
Tertile 2	1.48 (0.65–3.33)	1.27 (0.49–3.26)	1.23 (0.77–1.96)	0.99 (0.58–1.68)	1.62 (0.56–4.68)	0.90 (0.28–2.92)
Tertile 3	1.85 (0.84–4.10)	2.33 (0.93–5.83)	1.72 (1.09–2.70) *	1.53 (0.90–2.59)	1.40 (0.46–4.27)	1.23 (0.39–3.89)
	**Length at birth**					
	**Small**				**Large**	
	**OR (95%CI)**		**aOR (95%CI)**		**OR (95%CI)**	**aOR (95%CI)**
Tertile 1	1.00		1.00		1.00	1.00
Tertile 2	1.13 (0.63–2.01)		1.28 (0.64–2.56)		0.50 (0.05–5.60)	0.46 (0.04–5.47)
Tertile 3	1.17 (0.66–2.09)		1.56 (0.78–3.14)		1.05 (0.15–7.52)	0.42 (0.04–4.93)

UPF, ultra-processed foods; OR, odds ratio; aOR, adjusted odds ratio; 95%CI, 95% Confidence interval; OR adjusted by per capita income; maternal age; gestational weight gain; parity; physical activity; number of prenatal consultations. For all variables, “adequate” as reference. * Statistical significance.

## Data Availability

Not applicable.

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
