# Peer review of "Effect of Ultra-Processed Foods Consumption and Some Lifestyle Factors during Pregnancy on Baby’s Anthropometric Measurements at Birth"

_nutrients, 2022, doi:10.3390/nu15010044_

Round 1

Reviewer 1 Report

Review of Nutrients-2027714  

Title: Ultra-processed foods consumption during pregnancy and influence on baby's anthropometric measurements at birth                 

The authors took up an interesting topic about the relationship between ultra-processed foods consumption during pregnancy and baby's anthropometric measurements at birth                 

The title does not correspond to the content of the article. The authors analyzed not only the relationship between UPF supply but also socio-demographic, pregnancy and environmental factors. Therefore, I suggest changing the title to " Effect of ultra-processed foods consumption and some livestyle factors during pregnancy on baby's anthropometric measurements at birth "                               

The Introduction section is too modest. There is a lack of information on nutritional and non-nutritional factors influencing the anthropometric indicators of newborns. The authors also did not comprehensively demonstrate why UPF consumption would affect these indicators. Referencing the NOVA classification also requires a description in the Introduction section.               

The Material and Methods section lacks information on how many women participated in the study. The authors should supplement the data with the study flowchart. The authors used the FFQ questionnaire - it usually refers to consumption from the last year, the authors should specify the time range asked by the respondents.

In Table 1, the size of the group is not 626 for all parameters - why?            

References: 6, 8, 10, 16-18 in the text do not have brackets. The manuscript needs minor revision.

Author Response

Belo Horizonte, December 06, 2022

Dear Editors and Reviewers of Nutrients

Subject: Response to the changes suggested by the Editing Committee for the article entitled: “Ultra-processed foods consumption during pregnancy and influence on baby's anthropometric measurements at birth” – Manuscript ID: 2027714

Dear Editors, we appreciate for considering this work for publication and we respectfully present the changes that have been complied with and clarify the reviewers' doubts about the article. In this way, we hope to meet the expectations of this renowned journal.

Reviewer 1

Title: Ultra-processed foods consumption during pregnancy and influence on baby's anthropometric measurements at birth                 

The authors took up an interesting topic about the relationship between ultra-processed foods consumption during pregnancy and baby's anthropometric measurements at birth     

Answer: We appreciate your attention when considering our work. All points placed were carefully reviewed and we hope to meet the journal’s expectations. In addition, a rereading and adaptation of all topics requested by the reviewers was carried out.

Suggestion 1: The title does not correspond to the content of the article. The authors analyzed not only the relationship between UPF supply but also socio-demographic, pregnancy and environmental factors. Therefore, I suggest changing the title to " Effect of ultra-processed foods consumption and some lifestyle factors during pregnancy on baby's anthropometric measurements at birth”.

Answer: The change was made as suggested.

Suggestion 2: The Introduction section is too modest. There is a lack of information on nutritional and non-nutritional factors influencing the anthropometric indicators of newborns. The authors also did not comprehensively demonstrate why UPF consumption would affect these indicators. Referencing the NOVA classification also requires a description in the Introduction section. 

Answer: The entire section has been revised and rewritten.

Suggestion 3: The Material and Method section lacks information on how many women participated in the study. The authors should supplement the data with the study flowchart. The authors used the FFQ questionnaire - it usually refers to consumption from the last year, the authors should specify the time range asked by the respondents.

Answer: We added a figure on supplementary material.

The data was also added on the second paragraph of material and methods section.

The questionnaire consisted of 57 items with their respective frequencies of food consumption during pregnancy (last nine months) and quantifications expressed in home-made measurements.

Suggestion 4: In Table 1, the size of the group is not 626 for all parameters – why?

Answer: During data collection, we had some "missing values" when filling out the questionnaire and therefore not all sociodemographic variables could be evaluated in their entirety (n=626).

Information’s about to "missing data" was included in the footnotes of Table 1.

Suggestion 5: References: 6, 8, 10, 16-18 in the text do not have brackets. The manuscript needs minor revision.

Answer: The change was made as suggested.

Regards,

Rafaela Cristina Vieira e Souza

Reviewer 2 Report

Thank you for giving me the opportunity for review the manuscript entitled “Ultra-processed foods consumption during pregnancy and influence on baby's anthropometric measurements at birth”.

The manuscript is interesting and in scope of the Journal however it requires some clarifications.

The topic is of interest, as pregnant women have special dietary requirements that need to be met to prevent damage to the growing fetus as well as to prevent the development of certain diseases later in live.

Please find the specific comments below:

1.      In the introductory section of the manuscript, the health consequences for the mother, fetus and child associated with the mother's diet during pregnancy should be presented, and it would also be good to mention neurodevelopmental outcomes as long-term consequences of the prenatal diet.

2.  In the introductory section of the manuscript, more space should be devoted to the impact of poor maternal nutrition on the course and outcome of pregnancy

3.    One of the aim/advantage of this study proposed by authors, namely: “this study aimed to assess the influence of UPF consumption during pregnancy on baby’s anthropometric measurements at birth and at six months’’ cannot be achieved by this project as in this study there is no data (and analyses) for 6 months.

4.      It should be also more strongly underlined that  over or underreporting of nutrients intake could occur when the data are collected by the questionnaire  and that the declared food intake may also not correspond to the nutritional status measured by laboratory means (the concentration in blood)  as a result of different bioavailability of nutrients from different food products and individual differences in metabolism.

5. A strong limitation of this study is the lack of information on supplements/vitamins during pregnancy, the consumption of which is common and may affect the interpretation of the results.

6.     The results are poorly described, no information in main text included in the tables. Tables are just a supporting element and key information should be in the main text.

7.    In the supplementary section, there should (if possible) be a table of the percentage distribution of UPF.

Author Response

Belo Horizonte, December 06, 2022

Dear Editors and Reviewers of Nutrients

Subject: Response to the changes suggested by the Editing Committee for the article entitled: “Ultra-processed foods consumption during pregnancy and influence on baby's anthropometric measurements at birth” – Manuscript ID: 2027714

Dear Editors, we appreciate for considering this work for publication and we respectfully present the changes that have been complied with and clarify the reviewers' doubts about the article. In this way, we hope to meet the expectations of this renowned journal.

Reviewer 2

Thank you for giving me the opportunity for review the manuscript entitled “Ultra-processed foods consumption during pregnancy and influence on baby's anthropometric measurements at birth”.

The manuscript is interesting and in scope of the Journal however it requires some clarifications.

The topic is of interest, as pregnant women have special dietary requirements that need to be met to prevent damage to the growing fetus as well as to prevent the development of certain diseases later in live.

Please find the specific comments below

Answer: We appreciate your attention when considering our work. All points placed were carefully reviewed and we hope to meet the journal’s expectations. In addition, a rereading and adaptation of all topics requested by the reviewers was carried out.

Suggestion 1: In the introductory section of the manuscript, the health consequences for the mother, fetus and child associated with the mother's diet during pregnancy should be presented, and it would also be good to mention neurodevelopmental outcomes as long-term consequences of the prenatal diet.

Answer: The entire section has been revised and rewritten.

Suggestion 2: In the introductory section of the manuscript, more space should be devoted to the impact of poor maternal nutrition on the course and outcome of pregnancy

Answer: The entire section has been revised and rewritten.

Suggestion 3: One of the aim/advantage of this study proposed by authors, namely: “this study aimed to assess the influence of UPF consumption during pregnancy on baby’s anthropometric measurements at birth and at six months’’ cannot be achieved by this project as in this study there is no data (and analyses) for 6 months.

Answer: The change was made as suggested, and six months was removed from the aim.  

“… this study aimed to assess the influence of UPF consumption and some lifestyle factors during pregnancy on baby’s anthropometric measurements at birth.”              

Suggestion 4: It should be also more strongly underlined that over or underreporting of nutrients intake could occur when the data are collected by the questionnaire and that the declared food intake may also not correspond to the nutritional status measured by laboratory means (the concentration in blood) as a result of different bioavailability of nutrients from different food products and individual differences in metabolism.

Answer: We agree with your point of view. This information was added in the sixth paragraph of discussion.

“Another point that deserves attention and could explain the absence of associations from UPF intake and outcomes is that underreporting of food intake could occurs when the data are collected using FFQ [29], despite a validated questionnaire was used. Moreover, the declared food intake may also not correspond to the nutritional status measured by concentration in blood (blood biomarkers) as a result of different bioavailability of each nutrient and individual differences in metabolism [30].”

Suggestion 5: A strong limitation of this study is the lack of information on supplements/vitamins during pregnancy, the consumption of which is common and may affect the interpretation of the results.

Answer: During data collection from some studies previously carried out by our group, it was possible to identify that postpartum women do not always remember the dosage, frequency and period of supplements used during pregnancy. We asked all these items (dosage, frequency and period of use) and most mothers did not remember these data, however they remembered the name of the supplement, which we inserted in the first paragraph of the results: “Regarding supplementation, 88.5% of women used during pregnancy, the main ones re-ported being folic acid, iron and multivitamins”.

It is important to note that the nutrients commonly supplemented during pregnancy were not associated with the outcome. Another issue is that some studies also show that during the gestational period, women tend not to take the supplements as recommended, as there are several associated side effects, such as nausea, vomiting and gastrointestinal symptoms (Haider e Bhutta, 2017).

Additional reference: Haider BA, Bhutta ZA (2017) Multiple-micronutrient supplementation for women during pregnancy. Cochrane Database Syst Ver 4(4):CD004905. https://doi.org/10.1002/14651858

Suggestion 6: The results are poorly described, no information in main text included in the tables. Tables are just a supporting element and key information should be in the main text.

Answer: We added more information in the last two paragraphs of results.

“In the bivariate analysis, marital status, pre-gestational BMI, GWG, prenatal consultations and smoking was associated with baby’s weight at birth. The variables GWG and parity was associated with weight for gestational age at birth. And GWG was associated with length at birth. No association was observed between the UPF tertiles and the baby’s anthropometric measurements at birth. (Table 1).

In the regression model, before adjustment, there was a greater chance of insufficient weight at birth among babies of mothers in the highest tertile of UPF consumption. How-ever, such association was not observed after adjustments (Table 2)”.

Suggestion 7: In the supplementary section, there should (if possible) be a table of the percentage distribution of UPF.

Answer: We added a supplementary table with this information.

TABLE S1 - Mean maternal energy percentual consumption by food processing groups during pregnancy

Food processing group

% of total daily caloric value

In natura or minimally processed

50,01

Processed

19,43

Ultra-processed

30,56

Regards,

Rafaela Cristina Vieira e Souza

Round 2

Reviewer 2 Report

The authors have taken into account all the reviewer's comments. The manuscript has been revised correctly. Accepts in its present form.